# Common Molecular Targets of a Quinolone Based Bumped Kinase Inhibitor in *Neospora caninum* and *Danio rerio*

**DOI:** 10.3390/ijms23042381

**Published:** 2022-02-21

**Authors:** Joachim Müller, Nicoleta Anghel, Dennis Imhof, Kai Hänggeli, Anne-Christine Uldry, Sophie Braga-Lagache, Manfred Heller, Kayode K. Ojo, Luis-Miguel Ortega-Mora, Wesley C. Van Voorhis, Andrew Hemphill

**Affiliations:** 1Institute of Parasitology, Vetsuisse Faculty, University of Bern, Länggass-Strasse 122, 3012 Bern, Switzerland; joachim.mueller@vetsuisse.unibe.ch (J.M.); nocileta@gmail.com (N.A.); dennis.imhof@vetsuisse.unibe.ch (D.I.); kai.haenggeli@vetsuisse.unibe.ch (K.H.); 2Graduate School for Cellular and Biomedical Sciences, University of Bern, Mittelstrasse 43, 3012 Bern, Switzerland; 3Proteomics and Mass Spectrometry Core Facility, Department for BioMedical Research (DBMR), University of Bern, 3008 Bern, Switzerland; anne-christine.uldry@dbmr.unibe.ch (A.-C.U.); sophie.lagache@dbmr.unibe.ch (S.B.-L.); manfred.heller@dbmr.unibe.ch (M.H.); 4Center for Emerging and Re-Emerging Infectious Diseases (CERID), Division of Allergy and Infectious Diseases, Department of Medicine, University of Washington, Seattle, WA 98109, USA; ojo67kk@yahoo.ca (K.K.O.); wesley@uw.edu (W.C.V.V.); 5SALUVET, Animal Health Department, Faculty of Veterinary Sciences, Complutense University of Madrid, 28040 Madrid, Spain; luis.ortega@ucm.es

**Keywords:** affinity chromatography, binding proteins, proteomics, side effects, splicing

## Abstract

*Neospora caninum* is an apicomplexan parasite closely related to *Toxoplasma gondii*, and causes abortions, stillbirths and/or fetal malformations in livestock. Target-based drug development has led to the synthesis of calcium-dependent protein kinase 1 inhibitors, collectively named bumped kinase inhibitors (BKIs). Previous studies have shown that several BKIs have excellent efficacy against neosporosis in vitro and in vivo. However, several members of this class of compounds impair fertility in pregnant mouse models and cause embryonic malformation in a zebrafish (*Danio rerio*) model. Similar to the first-generation antiprotozoal drug quinine, some BKIs have a quinoline core structure. To identify common targets in both organisms, we performed differential affinity chromatography with cell-free extracts from *N. caninum* tachyzoites and *D. rerio* embryos using the 5-aminopyrazole-4-carboxamide (AC) compound BKI-1748 and quinine columns coupled to epoxy-activated sepharose followed by mass spectrometry. BKI-binding proteins of interest were identified in eluates from columns coupled to BKI-1748, or in eluates from BKI-1748 as well as quinine columns. In *N. caninum*, 12 proteins were bound specifically to BKI-1748 alone, and 105 proteins, including NcCDPK1, were bound to both BKI-1748 and quinine. For *D. rerio*, the corresponding numbers were 13 and 98 binding proteins, respectively. In both organisms, a majority of BKI-1748 binding proteins was involved in RNA binding and modification, in particular, splicing. Moreover, both datasets contained proteins involved in DNA binding or modification and key steps of intermediate metabolism. These results suggest that BKI-1748 interacts with not only specific targets in apicomplexans, such as CDPK1, but also with targets in other eukaryotes, which are involved in common, essential pathways.

## 1. Introduction

Apicomplexan parasites, in particular, the genera *Plasmodium*, *Cryptosporidium*, *Eimeria*, *Babesia*, *Theileria*, *Sarcocystis*, *Toxoplasma*, *Besnoitia* and *Neospora*, are pathogens causing serious disease in humans as well as in animals. In cattle, sheep or other ruminants, *Neospora caninum*, closely related to *Toxoplasma gondii*, causes abortion, stillbirths or birth of weak offspring [1,2]. Since neither vaccines nor therapeutic agents against neosporosis are currently available [3,4], there is an increasing interest in novel compounds potentially useful for the treatment of this disease.

Novel chemotherapies against neosporosis could be generated via the repurposing of known drugs, e.g., against malaria [5,6,7] or by de novo target-based drug development. An example of such targets are calcium-dependent protein kinases (CDPKs) with homologs in plants, apicomplexans and other phyla, but not in animals. Consequently, during the last decade, CDPKs, in particular, CDPK1, have been intensively investigated as targets for drug development against *Toxoplasma gondii* [8] and other related apicomplexans [9]. These studies have led to the development of bumped kinase inhibitors (BKIs) of CDPK1 [10,11]. CDPK1 is crucially involved in invasion of host cells and egress [12].

In studies using the mouse model, the pyrazolopyrimidine (PP) BKI-1294 has yielded promising results against vertical transmission of neosporosis [13,14] and toxoplasmosis [15] in mice, as well as against toxoplasmosis in sheep [16]. Another PP compound, BKI-1553, was also shown to exhibit protective effects in mice and inhibited vertical transmission and abortion in *N. caninum* infected pregnant sheep [17]. Other BKIs were, however, less promising. In particular, the 5-aminopyrazole-4-carboxamide (AC) BKI-1517 showed good therapeutic efficacy against neosporosis in adult animals and protection from vertical transmission but was shown to have detrimental effects on fertility in mice [18]. Another BKI with promising efficacy in vitro and in vivo is the AC compound BKI-1748 [19]. This BKI had a profound impact on vertical transmission of *N. caninum* in pregnant mice when applied at 20 mg/kg/day for five days, but fertility in mice was dramatically reduced when applied at 50 mg/kg/day for five days.

In vitro studies using *N. caninum* infected fibroblasts showed that several BKIs, including BKI-1294, BKI-1553 and BKI-1748, did not act parasiticidally. Rather, exposure of intracellular tachyzoites to these compounds induced the formation of multinucleated complexes, which were composed of newly formed zoites devoid of the outer plasma membrane, and were incapable of separating from each other and forming infective tachyzoites [13,14,19]. These findings suggest that BKIs could act on other targets besides CDPK1.

In addition, several BKIs, including BKI-1748, were shown to cause embryonic malformations in a zebrafish (*Danio rerio*) model when applied during the first 96 h post-fertilization at concentrations of 20 µM and above [20]. BKI-1517 and BKI-1553 also exerted embryonic malformations in zebrafish, although already at 0.2 and 5 µM, respectively [20]. These BKIs share a quinoline core (see structures in [20]), which may at least be partially responsible for the side effects of these compounds. Quinine, one of the first antiprotozoal drugs, has long been known to affect pregnancy [21,22], as discussed elsewhere [23]. Consequently, we hypothesize that BKI-1748 could interact with common targets in apicomplexans and animals, and that this quinoline motif could play a role. To investigate our hypothesis, we performed differential affinity chromatography followed by mass spectrometry with BKI-1748 and quinine (Figure 1) on cell-free extracts from *N. caninum* tachyzoites and *D. rerio* embryos.

Here, we present differential affinity chromatography (DAC) results showing that common BKI-1748 binding proteins were identified in both organisms.

## 2. Results

### 2.1. Scanning Electron Microscopy (SEM) of Neospora caninum and Danio rerio Treated with BKI-1748

When human foreskin fibroblasts were infected with *N. caninum* tachyzoites in the absence of BKI-1748, numerous tachyzoites were formed and egressed from the host cells. In the presence of BKI-1748, parasites formed larger complexes that had to be physically released from their host cells in order to visualize them for SEM (Figure 2).

After 96 h post-fertilization, untreated *D. rerio* embryos exhibited a normal phenotype, as expected for this stage of development, as depicted in Figure 3.

When treated with BKI-1748, *D. rerio* embryos showed distinct malformations, such as an enlarged thorax (at 20 µM) or even a burst of the thorax (at 40 µM). Moreover, the eye lost its characteristic morphological appearance upon BKI-treatment (Figure 4.)

### 2.2. DAC Proteomes of Neospora caninum and Danio rerio

Mass spectrometry analysis of the proteomes obtained after DAC of cell-free extracts of *N. caninum* tachyzoites resulted in the identification of 8089 unique peptides matching to 1162 proteins. The complete dataset is given in Appendix A, which is available online. The corresponding numbers for the proteomes obtained by DAC of *D. rerio* embryos were 9381 unique peptides matching to 1202 proteins. The complete dataset is given in Appendix A. In both datasets, the protein intensity distributions were almost equal for all column eluates (Figure 5).

Unbiased analysis of the dataset by principal component analysis (PCA) demonstrated that the differential affinoproteomes eluted from mock, quinine and BKI-1748 columns were located in non-overlapping clusters separated by both principal components. This was true for both organisms and with both Top3 and LFQ algorithms (Figure 6).

After the subtraction of proteins binding to the mock columns, the affinoproteome of *N. caninum* comprised 424 proteins specifically binding to BKI-1748 or quinine columns or both, the affinoproteome of *D. rerio* comprised 357 proteins, as depicted in Figure 7.

### 2.3. Specific BKI-1748 Binding Proteins in N. caninum and D. rerio

As shown in Figure 7, only 12 *N. caninum* proteins could be identified to be binding specifically to BKI-1748 (Table 1). NcCDPK1 (NCLIV_011980), the target, for which BKI-1748 and related compounds had been designed, was not identified as a protein specifically binding to BKI-1748 but was also detected in eluates from mock and quinine columns. The respective LFQ values were 22 × 10^6^ (BKI-1748), 4 × 10^6^ (quinine) and 9 × 10^6^ (mock; Appendix A).

By far, the most abundant BKI-1748 binding protein is a 260 amino acid hypothetical protein encoded by ORF NcLIV_020480 (Figure 8A). The closest homologs are U1 70 kDa small nuclear ribonucleoproteins conserved amongst various phyla. Since the protein encoded by ORF NcLIV_020480 is much shorter, the homologies only extend to the N-terminal sequences of these ribonucleoproteins (Figure 8B).

The closest homolog for structural modeling was the template 6qx9.6.A with 51.30% sequence identity corresponding to the U1 small nuclear ribonucleoprotein 70 kDa from the structure of a human fully-assembled precatalytic spliceosome pre-B complex [24], presented in Figure 8C.

Interestingly, the second most abundant protein NCLIV_069960 encoding an RNA-binding protein, and two other BKI-1748 binding proteins, an RNA polymerase, and a t-RNA-synthase, were related to transcriptional and translational processes.

Moreover, alveolin1, member of a family of proteins uniquely found in the phylum Alveolata [25], was amongst the proteins binding specifically to BKI-1748.

The *D. rerio* pull-down yielded 13 proteins specifically binding to BKI-1748. By far, the most abundant protein was A0A140LGT9_DANRE encoding a 155 amino acid protein with a predicted signal peptide at its N-terminus (amino acids 1–22). The processed peptide chain is homologous to leucocyte-derived chemotaxin 2 from other teleost species; the second most abundant binding protein was syntenin (Table 2).

Moreover, proteins involved in RNA binding and modification, namely splicing factor 1, a microsomal ribsosomal protein, and a pre-mRNA processing factor were identified.

### 2.4. Proteins Binding to Both BKI-1748 and Quinine Columns

Besides proteins binding specifically to BKI-1748, proteins binding to both BKI-1748 and quinine columns are of interest. In eluates from *N. caninum* tachyzoites, 105 proteins binding to both columns were identified (Appendix A). Within this subset, the most abundant protein was the 180 amino acid polypeptide encoded by ORF NCLIV_068520, annotated as a MIC5 homolog by the ToxoDB. In *T. gondii*, the microneme protein MIC5 is secreted onto the parasite surface during invasion and acts as an inhibitor of the proteolytic processing of microneme and dense granule proteins secreted by tachyzoites, by interfering in the functional activity of subtilisin 1 [26]. It is conceivable that similar mechanisms are also important for the invasion process of *N. caninum* tachyzoites. Moreover, hypothetical proteins without annotated putative functions were identified, and six of the 20 most abundant proteins appear to be involved in transcription and/or translation (Table 3).

In eluates from *D. rerio* embryos, 98 proteins binding to both columns were identified (Appendix A). Within this subset, the most abundant protein was the vitellogenin 1 homolog encoded by Q8JH37_DANRE, followed by a lysyl endopeptidase and by another vitellogenin 1 homolog. Besides proteins with diverse functions, six of the 20 most abundant proteins were involved in transcription and/or translation (Table 4).

When looking at the specific BKI-1748 binding proteins with proteins binding to BKI-1748 as well as to quinine and evaluating their potential functions in both *N. caninum* and *D. rerio*, it was evident that most drug-binding proteins were related to RNA-binding and modification, in particular to RNA splicing, namely 20 of 117 proteins for *N. caninum* and 37 of 111 proteins for *D. rerio*. The second most prominent group consisted of proteins binding to and/or modifying other proteins such as peptidases or chaperonins involved in protein refolding with 17 proteins in each group (Table 5).

## 3. Discussion

When interpreting differential affinity chromatography data, care must be taken not to underestimate the number of proteins binding to a peculiar compound. If we look at CDPK1, the target enzyme for which BKI-1748 has been created, we see that this protein was identified not only in eluates from BKI-1748, but also from quinine and mock columns. This may be explained by interactions with proteins that bind to the mock columns in an unspecific manner. In a co-immunoprecipitation study using an anti-CDPK1 antibody, we have identified NcMIC3 as a major interaction partner of NcCDPK1, indicating that this kinase may form complexes with other proteins (unpublished data). This protein is found in all eluates, including the mock column (Appendix A). The number of BKI-binding proteins may therefore be higher than the ones we have seen in our eluates. In all likelihood, the proteins specifically identified in BKI-1748 eluates interacted with the non-quinoline moiety of this molecule only, whereas the proteins identified in both BKI-1748 and quinine eluates may have bound to the quinoline moiety.

Considering the proteins binding specifically to BKI-1748, we identified organism specific subsets, such as hypothetical proteins or alveolin 1 in *N. caninum*, and proteins specifically involved in development, such as chemotaxin 2 or vitellogenin in *D. rerio.*

Alveolin 1 is a member of the alveolin family, a class of proteins unique to the phylum alveolata. These proteins are intermediate filament-like proteins and are associated to alveolar sacs located beneath the plasma membrane forming the inner membrane complex, which interacts with the actin–myosin system playing pivotal roles in cell division [27] and motility [28]. As cellular motility in *N. caninum* is closely linked to host cell invasion, the binding of BKI-1748 to this structural element could contribute to the pronounced inhibition of invasion seen during BKI-treatments. Intracellular tachyzoites that are exposed to BKI-1748 form multinuclear complexes that contain numerous newly formed zoites, which cannot complete the last steps in cytokinesis, thus parasites remain trapped within their host cells [19]. Interference with cell division by binding to alveolin could be partially responsible for this effect.

In *D. rerio*, a protein with similar functions, namely syntenin, is the second most abundant BKI-1748 specific binding protein. Similar to alveolins, this protein is associated with membrane vesicles, in particular, in neuronal cells, and is involved in cell proliferation, excretion and motility [29]. In zebrafish embryo development, the protein is involved in the spreading and the thinning of the blastoderm closure of the blastopore. Syntenin interacts with syndecan heparan sulphate proteoglycans, which act as co-receptors for adhesion molecules and growth factors. In addition, syntenin binds to phosphatidylinositol 4,5-bisphosphate and with the small GTPase ADP-ribosylation factor 6, which regulate the endocytic recycling of syndecan. Earlier studies have proposed a role for syntenin in the autonomous vegetal expansion of the yolk syncytial layer and the rearrangement of the actin cytoskeleton in extra-embryonic tissues, but not in embryonic cell fate determination [30].

The most abundant BKI-1748 binding protein in *D. rerio* is chemotaxin 2, which is homologous to mouse leukocyte cell-derived chemotaxin 2 (LECT2). LECT2 was initially isolated as a possible chemotactic factor for neutrophils [31] and has subsequently been demonstrated to be involved in various processes such as liver regeneration and natural killer cell development [32], carcinogenesis [33], and anti-infective and anti-inflammatory responses [34]. However, its actual function in zebrafish embryos remains obscure.

Moreover, confirming our initial hypothesis that common targets may exist in both organisms, we have identified BKI-1748 binding proteins involved in various common pathways with predominant subsets of RNA binding and modifying proteins. Interestingly, the most abundant BKI-1748 specific binding protein was the uridine-rich small ribonuclear binding protein U1 (NcsnrpU1), an essential component of the spliceosome [24]. U1 recognizes specific sequence motifs at the intron–exon boundaries. This would allow generating knockdown mutants by inserting recognition sequences at the 3′-ends of exons of genes of interest [35], thereby validating the function of this protein for apicomplexans. Since the mechanisms of RNA splicing are highly conserved amongst eukaryotes, it is not surprising to find two U1 homologs in the BKI-1748 binding protein fraction from *D. rerio*, namely, E7F071_DANRE and Q6NUT5_DANRE. These homologs were, however, much longer than NcsnrpU1 with only 44 identical and 76 similar amino acid positions. The homolog with the highest identity to NcsnrpU1, Q4KMD3, is not found in the eluates of any column (Appendix A). Moreover, the small ribonucleoprotein Sm is another example found in both binding protein subsets. A list of BKI-1748 binding proteins involved in RNA splicing is shown in Table 6.

This list may not be complete since in the case of *N. caninum*, many proteins are not annotated. However, it underlines the impact BKI-1748 (and perhaps compounds with similar structures) could potentially have on RNA splicing. Since RNA splicing is of major importance in embryonic development, interference with this pathway, via mutations of the components or via interaction with chemotherapeutics, may explain the pleiotropic phenotypes observed in *D. rerio* development, such as the eye and thorax malformations we have observed, and the impact on fertility in pregnant mouse models.

Together, target-based drug development is not a highway to safe drugs without side effects. Off-target effects have to be evaluated for any novel compound using appropriate models. The identification of drug-binding proteins via DAC—as exemplified in the present study—may constitute a rapid way to gain insights into the complex interactions that take place. While it is important to keep in mind that the binding of a drug to a given protein does not automatically imply functional inhibition, DAC still has considerable potential to eliminate undesired side effects by optimizing compounds for the function of their affinoproteomes.

## 4. Materials and Methods

### 4.1. Tissue Culture Media, Biochemicals and Compounds

Cell culture media were purchased from Gibco-BRL (Zürich, Switzerland), and biochemical agents were procured from Sigma St. Louis, MO, USA).

### 4.2. In Vitro Culture and Collection of Parasites

For affinity chromatography necessitating high yields, *Neospora caninum* NcSpain-7 tachyzoites were grown in HFF as previously described [43,44]. The pellets of tachyzoites were stored at −80 °C until processing.

### 4.3. Maintenance and Collection of Zebrafish Embryos

Zebrafish embryos were kindly provided by the group of Nadia Mercader, from the Institute of Anatomy, University of Bern. Fertilized zebrafish embryos were maintained at 28 °C in 50 mL falcon tubes containing E3 medium (5 mM NaCl, 0.17 mM KCl, 0.33 mM CaCl_2_ and 0.33 mM Mg SO_4_, buffered to pH 7.5 using sodium bicarbonate) during 24 h. At 24 h post-fertilization, all embryos were euthanized by immersion in a solution of pre-cooled 3-101 aminobenzoic acid ethyl ester (100 μg/L; MS222; Argent Chemical Laboratories, Redmond, WA, USA), centrifuged and placed at −20 °C [20].

### 4.4. Scanning Electron Microscopy (SEM)

HFF monolayers grown in T25 tissue culture flasks were infected with 1 × 10^7^ NcSpain-7 tachyzoites in culture medium (4 h, 37 °C/5% CO_2_). Then, cultures underwent continuous treatment with 2.5 μM BKI-1748 for 4 days. Subsequently, MNCs and tachyzoites were separated from host cell debris by Sephadex-G-25-chromatography and were fixed and processed as described [19].

Fertilized zebrafish embryos were maintained for 48 h allowing the embryos to emerge (change of medium after 24 h) as described above. Subsequently, they were exposed to either 10, 20 or 40 µM of BKI-1748, or were left untreated, during another 48 h. At 96 h post-fertilization, embryos were euthanized as above, and fixed and processed for SEM as described [19].

All specimens were placed onto glass coverslips, sputter-coated with gold and inspected on a Zeiss Gemini450 SEM operating at 5 kV.

### 4.5. Protein Extraction and Affinity Chromatography

For protein extraction, frozen pellets of *N. caninum* Sp7 tachyzoites and zebrafish embryos were resuspended in ice-cold extraction buffer, i.e., PBS containing 1% Triton X-100 and 1% of Halt proteinase inhibitor cocktail (ThermoFisher). Suspensions were vortexed thoroughly and centrifuged (13,000× *g* rpm, 10 min, 4 °C). Extraction of pellets was repeated twice. Three ml of extraction buffer was used in total. Supernatants were combined (approximately 3 mg of total protein) and subjected to affinity chromatography.

In order to produce the sepharose matrices conjugated to BKI-1748 or quinine, 0.5 g lyophilized epoxy–sepharose with a C12 spacer was suspended in 15 mL H_2_O and centrifuged at 300× *g* for 5 min. Washes in water were repeated twice followed by a wash with coupling buffer (0.1 M NaHCO_3_, pH 9.5). After the last wash, 20 mg of each compound dissolved in 2.5 mL DMSO were added and coupling buffer was added to a maximum volume of 5 mL. Mock column medium was generated by incubating 0.5 g of epoxy–sepharose only with DMSO. The mixture was incubated for 3 days at 37 °C under slow but continuous shaking in order to allow coupling of the epoxy group to the compounds. The resulting column medium (approximately 2 mL) was washed with coupling buffer (15 mL) followed by a wash with ethanolamine (1 M, pH 9.5) and by an incubation in 10 mL ethanolamine for 4 h at 20 °C in the absence of light in order to block residual reactive groups. The column medium was then transferred to a chromatography column (Novagen, Merck, Darmstadt, Germany), and extensively washed with PBS-DMSO (1:1) and PBS in order to remove unbound compounds. The columns were stored in PBS containing 0.02% NaN_3_ at 4 °C.

Prior to affinity chromatography, mock columns were combined to either quinine or BKI-1748 columns in tandem (mock first, then compound) and washed with 50 mL PBS equilibrated at 20 °C. Crude extracts (3 mL), prepared as described above, were loaded with a flow rate of 0.25 mL/min. The column was washed with PBS until the baseline was flat (10 column volumes, corresponding to ca. 25 mL). Then, the columns were separated, and the binding proteins were eluted with 50 mM acetic acid (5 mL per column). The eluates were lyophilized and stored at −80 °C [45].

### 4.6. Proteomic Analysis of the Eluted Proteins by Mass Spectrometry

The lyophilized eluates were dissolved in 10 μL of 8 M urea and 0.1 M of Tris-HCl^-^ (pH = 8), then, 1 μL of 0.1 M of Tris-Cl^-^ (pH = 8) buffer containing 0.1 M of dithiothreitol were added, followed by incubation for 30 min at 37 °C and constant mixing with 600 rpm. This step was repeated with 1 μL of 0.5 M of iodoacetamide. Iodoacetamide was quenched by the addition of 5 μL 0.1 M of Tris-HCl^-^ (pH = 8) and the urea concentration further diluted to 4 M by the addition of 2 mM calcium dichloride in 20 mM Tris buffer. Proteins were digested for 2 h at 37 °C by the addition of 1 μL of 0.1 μg/μL LysC sequencing grade protease (Promega), followed by further dilution of urea to 1.6 M with above calcium dichloride buffer and 1 μL of 0.1 μg/μL trypsin sequencing grade (Promega). Digestion was completed by incubation over night at ambient room temperature. Digestion was stopped with 2.5 μL of 20% (*v*/*v*) trifluoroacetic acid. After an incubation for 15 min at room temperature, the digest was spun for 1 min at 16,000× *g*, and the cleared supernatant was transferred to a HPLC vial for subsequent nano-liquid reverse phase chromatography coupled to tandem mass spectrometry, as described earlier [46].

The mass spectrometry data were processed with MaxQuant (v1.6.14.0), as described earlier, against a current protein sequence database release from toxodb.org for *Neospora caninum* (ToxoDB-54_NcaninumLIV_AnnotatedProteins.fasta), and against Uniprot (www.uniprot.org, accessed on 9 December 2021) for *Danio rerio*. The MaxQuant search results were then further processed in R-studio. Imputation was not applied. Consequently, the output indicated only on–off proteins and labeled those proteins that appeared to be the most extreme outliers in a linear regression comparison. Alignments were performed using the ClustalO tool provided by the Expasy network (www.expasy.org, accessed on 9 December 2021). Modeling of the structure of a binding protein was performed using the Swiss Model homology modeling tools [47] of the Swiss Model repository [48], as provided by the Expasy network.

## Figures and Tables

**Figure 1 ijms-23-02381-f001:**
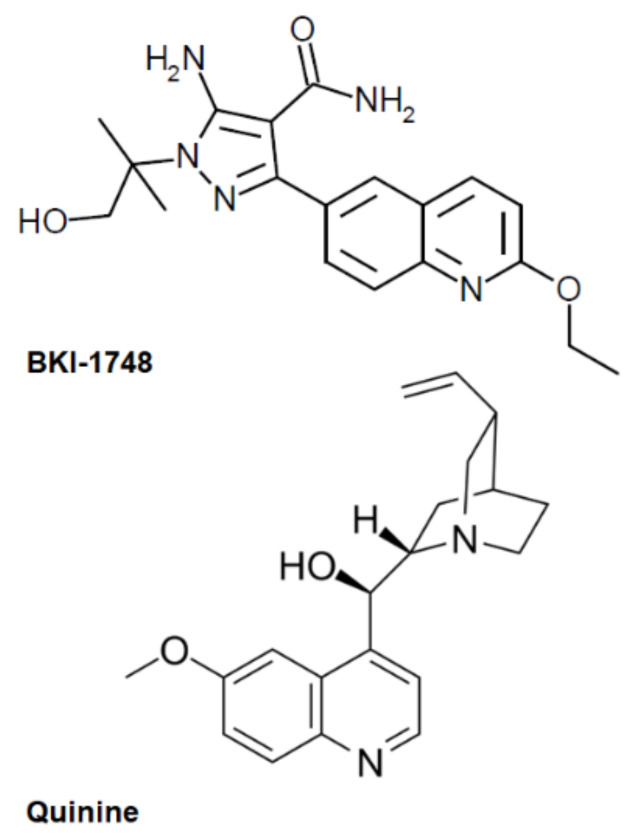
Structures of the compounds used for differential affinity chromatography.

**Figure 2 ijms-23-02381-f002:**
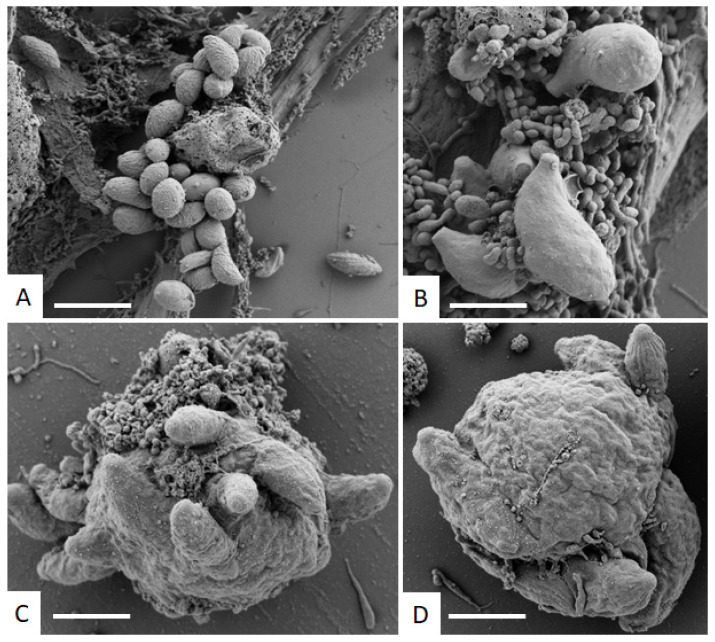
SEM of *N. caninum* tachyzoites cultured in HFF without (**A**,**B**) and with BKI-1748 (2.5 µM; (**C**,**D**)) for 4 days. Bars in **A** = 4.5 µm; **B** = 1.8 µm; **C** and **D** = 3 µm.

**Figure 3 ijms-23-02381-f003:**
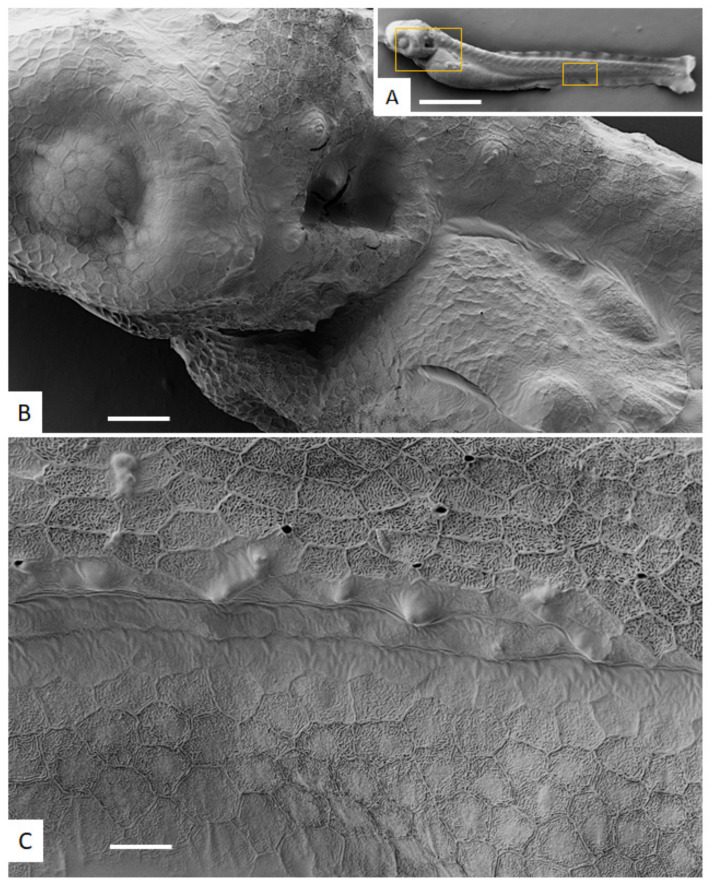
SEM of untreated zebrafish embryo at 96 h post-fertilization. (**A**) shows the entire embryo. The boxed area at the anterior part around the eye is magnified in (**B**), a portion of the skin in the midbody region is magnified in (**C**). Note the hexagonal pattern covering the entire embryo cuticle surface. Bars in **A** = 560 µm, **B** = 50 µm, in **C** = 25 µm.

**Figure 4 ijms-23-02381-f004:**
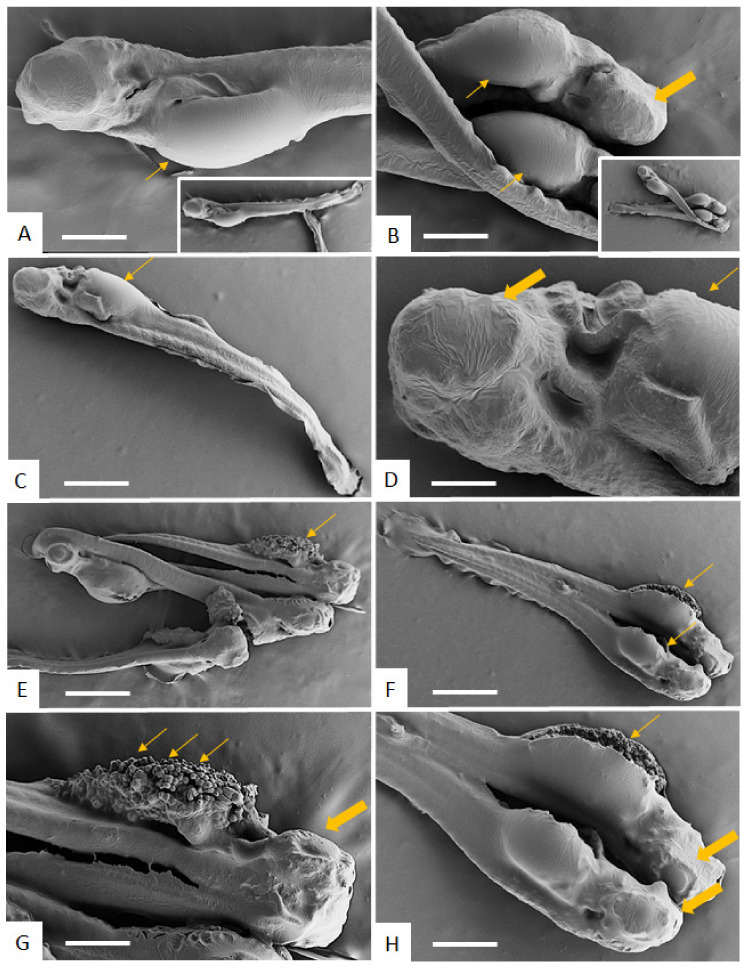
SEM of zebrafish embryos at 96 h post-fertilization treated with 20 µM (**A**–**D**) or 40 µM (**E**–**H**) BKI-1748. The inserts in (**A**,**B**) show low magnification views, and (**E**,**F**) are low magnification views of (**G**,**H**), respectively. Thin arrows point towards an enlarged thorax (20 µM, (**A**–**D**)), and treatment with 40 µM causes bursting of the thorax (**E**–**H**). Thicker arrows indicate malformation of the eye. Bars in **A** and **B** = 240 µm, **C** = 460 µm, **D** = 60 µm, **E** = 600 µm, **F** = 500 µm, **G** and **H** = 280 µm.

**Figure 5 ijms-23-02381-f005:**
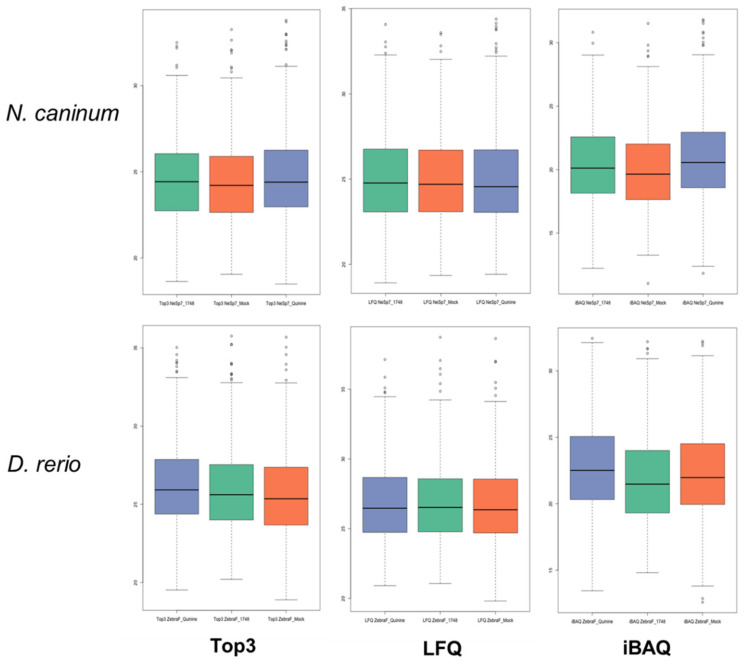
Protein intensity distributions of the proteome datasets presented by Appendix A (*N. caninum*) and Appendix A (*D. rerio*) as calculated by the Top3, LFQ and iBAQ algorithms. Cell-free extracts were prepared and subjected to differential affinity chromatography on mock (green), quinine (orange), or BKI-1748 (blue) columns followed by mass spectrometry as described in the Materials and Methods Section.

**Figure 6 ijms-23-02381-f006:**
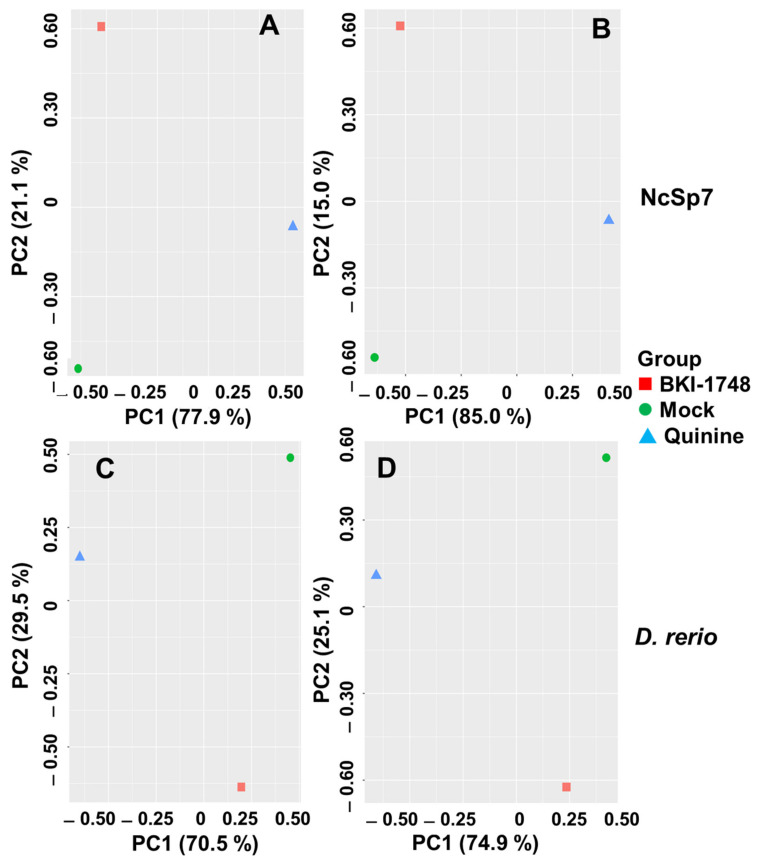
Principal component analysis of affinoproteome data sets from *N. caninum* tachyzoites (**A**,**B**) and from *D. rerio* embryos (**C**,**D**). Cell-free extracts were prepared and subjected to differential affinity chromatography on mock (green circle), quinine (blue triangle) or BKI-1748 (red square) columns followed by mass spectrometry as described in the Materials and Methods Section. (**A**,**C**) Top3 data; (**B**,**D**) LFQ data. X-axis, principal component 1; Y-axis, principal component 2.

**Figure 7 ijms-23-02381-f007:**
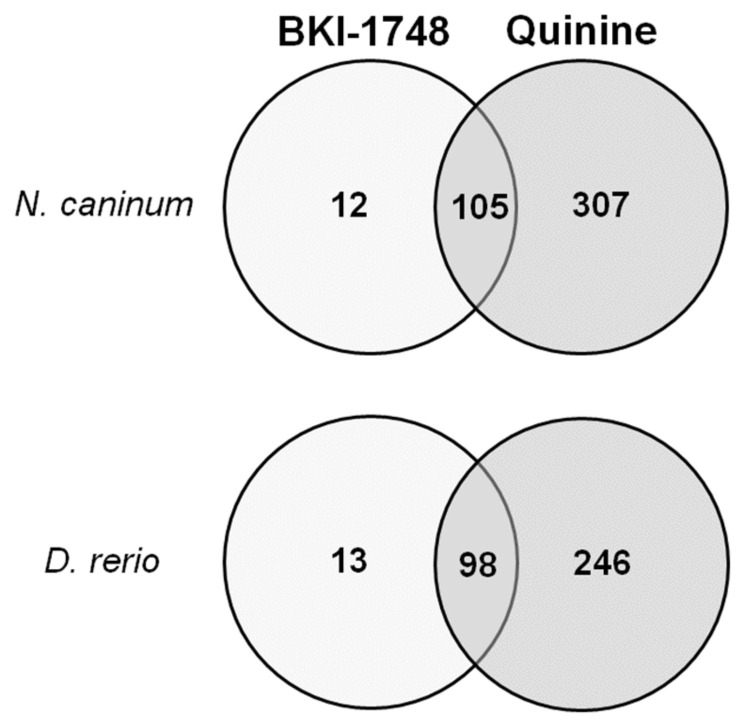
Venn diagram detailing the number of proteins identified by differential affinity chromatography. Eluates from *N. caninum* and *D. rerio* from BKI-1748 and quinine columns were compared by MS shotgun analysis as described in Materials and Methods.

**Figure 8 ijms-23-02381-f008:**
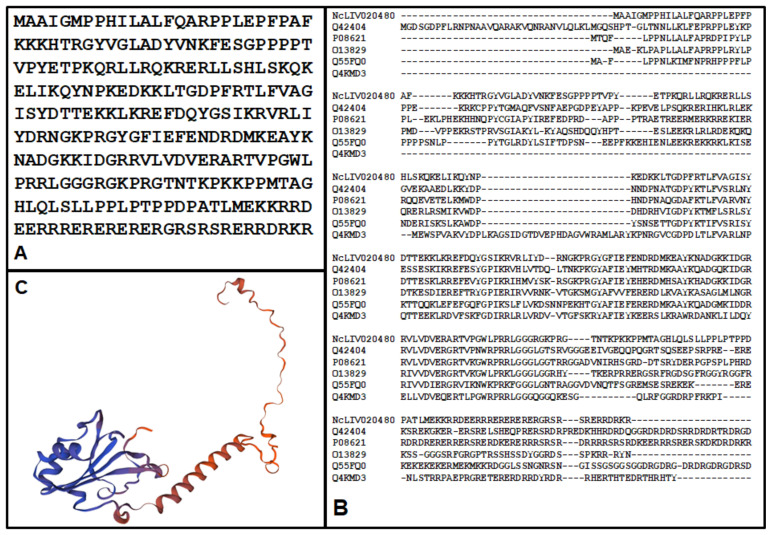
Primary sequence of ORF NcLIV_020480 (**A**), alignments to the N-terminal sequences of five homologs, i.e., Q42404.1 (*A. thaliana*), P08621 (*H. sapiens*), Q13829 (*S. pombe*), Q55FQ0 (*D. discoideum)* and Q4KMD3 (*D. rerio*) (**B**), and a model structure calculated by Swiss Model from the structure of the human homolog (6qx9.6.A) with hydrophilic residues in blue and hydrophobic residues in red (**C**).

**Table 1 ijms-23-02381-t001:** List of the 12 proteins from *Neospora caninum* specifically binding to BKI-1748, as identified by differential affinity chromatography followed by mass spectrometry. See Appendix A for the full dataset. The relative abundance (rAbu) is based on iBAQ, so that the sum or rAbu is 1,000,000 for every column. The ORF numbers from ToxoDB are provided.

ORF Number	Annotation	rAbu
NCLIV_020480	Hypothetical protein; U1 small nuclear ribonucleoprotein	435
NCLIV_069960	RNA binding motif-containing zinc finger protein, putative	110
NCLIV_046620	*Plasmodium vivax* PV1H14060_P, related	101
NCLIV_054840	Conserved hypothetical protein	98
NCLIV_039750	Hypothetical protein	81
NCLIV_026500	Alveolin1, related	50
NCLIV_070040	RNA polymerase II accessory factor Cdc73p	32
NCLIV_039960	Hypothetical protein	29
NCLIV_007730	Valyl-tRNA synthetase, mitochondrial, related	29
NCLIV_066970	Putative enoyl-acyl carrier reductase	17
NCLIV_051300	Nifu-pending protein, related	14
NCLIV_022400	Hypothetical protein	9

**Table 2 ijms-23-02381-t002:** List of the 13 proteins from *Danio rerio* embryos specifically binding to the BKI-1748 column, as identified by differential affinity chromatography followed by mass spectrometry. See Appendix A for the full dataset. The relative abundance (rAbu) based on iBAQ, so that the sum or rAbu is 1,000,000 for every column.

Uniprot Number	Annotation	rAbu
A0A140LGT9_DANRE	Si:ch211-132p1.4 leucocyte-derived chemotaxin 2 homolog	352
Q803J6_DANRE	Syndecan binding protein (syntenin)	211
Q804G3_DANRE	Annexin	70
A0A0R4IGU9_DANRE	LRP chaperone MESD	42
UB2V2_DANRE	Ubiquitin-conjugating enzyme E2 variant 2	23
Q6K199_DANRE	C-Related protein	21
F1QP64_DANRE	Splicing factor 1	20
Q7ZVS1_DANRE	Hematological and neurological expressed 1-like	15
A3KP51_DANRE	Mitochondrial ribosomal protein L37	13
Q7ZUE4_DANRE	PRP40 pre-mRNA processing factor 40 homolog A (Yeast)	9
F1QC72_DANRE	Alkylglycerone-phosphate synthase	7
Q6IQF7_DANRE	Ppig protein (Fragment)	7
A0A1L1QZF2_DANRE	Transaldolase	7

**Table 3 ijms-23-02381-t003:** List of the twenty most abundant proteins from *Neospora caninum* tachyzoites binding to both BKI-1748 and quinine columns, as identified by differential affinity chromatography followed by mass spectrometry. See Appendix A for the full dataset. The relative abundance (rAbu) is based on iBAQ, so that the sum or rAbu is 1,000,000 for every column. The proteins are listed according to their decreasing rAbu values in BKI-1748 eluates.

ORF Number	Annotation	rAbu BKI-1748	rAbu Quinine
NCLIV_068520	Unspecified product; MIC5 homolog	795	224
NCLIV_062730	Hypothetical protein; mitotic checkpoint protein, BUB3 family protein	228	16
NCLIV_021130	Conserved hypothetical protein	193	9
NCLIV_006770	Conserved hypothetical protein	175	11
NCLIV_033040	RNA binding motif 28 protein, related	146	609
NCLIV_057700	Hypothetical protein, Ribosomal L30 N-terminal domain.	134	146
NCLIV_061340	EF-1 guanine nucleotide exchange domain containing protein, putative	128	13
NCLIV_038830	Sm protein, related	97	50
NCLIV_019020	Conserved hypothetical protein	93	3
NCLIV_068400	Unspecified product, putative nucleoside-triphosphatase	66	391
NCLIV_007620	Putative replication protein A2	61	830
NCLIV_054170	RNA recognition motif-containing protein	58	2
NCLIV_059150	C2 domain containing protein, putative	56	10
NCLIV_048580	Putative RNA recognition motif 2 domain-containing protein	51	9
NCLIV_034160	Glutamine synthetase, related	50	26
NCLIV_045530	Conserved hypothetical protein, Tim10/DDP family zinc finger containing protein, putative	43	1232
NCLIV_011140	gl18351, related; putative TCP-1 chaperonin	42	39
NCLIV_034900	Putative casein kinase II beta chain	36	88
NCLIV_062570	Contig An13c0020, complete genome, related	31	27
NCLIV_037780	Putative zinc finger (CCCH type) protein	29	63

**Table 4 ijms-23-02381-t004:** List of the 20 most abundant proteins from *Danio rerio* embryos binding to both BKI-1748 and quinine columns, as identified by differential affinity chromatography followed by mass spectrometry. See Appendix A for the full dataset. The relative abundance (rAbu) is based on iBAQ, so that the sum or rAbu is 1,000,000 for every column. The proteins are listed according to their decreasing rAbu values in BKI-1748 eluates.

Accession Number	Annotation	rAbu BKI-1748	rAbu Quinine
Q8JH37_DANRE	Vitellogenin 1 (Fragment)	665.9	362.5
LYSC_LYSEN	Lysyl endopeptidase	628.7	81.2
Q8JH36_DANRE	Vitellogenin 1 (Fragment)	485.1	218.1
A8WG29_DANRE	LOC561392 protein	380.4	51.4
F6P0Y5_DANRE	Cytochrome b-c1 complex subunit 6	311.8	1309.6
F1RDV9_DANRE	Polymerase (RNA) II (DNA directed) polypeptide	289.5	3548.8
Q8JHH1_DANRE	Small nuclear ribonucleoprotein Sm D1	259.7	483.5
Q7ZW80_DANRE	Alpha-L-fucosidase	227.5	807.0
A4FVJ1_DANRE	Im:6908808 protein	138.5	10.7
A0A0R4ICY2_DANRE	RNA-binding motif protein 24a	88.8	155.5
PARK7_DANRE	Parkinson disease protein 7 homolog	65.1	22.9
E9QFI5_DANRE	Diazepam-binding inhibitor (GABA receptor modulator, acyl-CoA binding protein)	64.4	42.3
U3JAH9_DANRE	BUB3 mitotic checkpoint protein	61.5	3.1
Q6PBJ8_DANRE	Peptidylprolyl isomerase	60.9	21.3
F6P4Z6_DANRE	NADH:ubiquinone oxidoreductase subunit V3	52.8	151.1
Q1LWX4_DANRE	Proteasome subunit alpha type (Fragment)	49.8	30.9
Q8JHJ3_DANRE	U2 small nuclear RNA auxiliary factor small subunit	47.0	33.6
A8E587_DANRE	MGC174152 protein	47.0	61.4
F1QII8_DANRE	Eukaryotic translation initiation factor 3 subunit	38.1	179.3
RBM8A_DANRE	RNA-binding protein 8A	36.7	160.2

**Table 5 ijms-23-02381-t005:** Summary of putative functions of BKI-1748 and quinine-binding proteins in *Neospora caninum* and *Danio rerio*. The table comprises proteins only identified in BKI-1748 column eluates (see Table 1 and Table 2), as well as proteins identified in both BKI-1748 and quinine eluates (see Appendix A for complete list). The functions were identified based on information provided by Uniprot (www.uniprot.org, accessed on 10 November 2021) and related databases.

Function	*N. caninum*	*D. rerio*
RNA binding and modification	20	37
DNA binding and modification	3	14
Protein binding and modification	17	17
Cytoskeleton and intracellular transport	10	15
Intermediary metabolism	12	17
Ambiguous or hypothetical	55	4
Embryo specific	-	7
**Total**	**117**	**111**

**Table 6 ijms-23-02381-t006:** List of BKI-1748 binding proteins in *Neospora caninum* and *Danio rerio* with homologs involved in RNA splicing. The proteins are listed by decreasing abundance in the respective subsets. Snrp: small nuclear ribonucleoprotein; UP: information via Uniprot database (www.uniprot.org, accessed on 9 December 2021).

Protein	Accession N°	Function	Reference
*N. caninum* snrpU1	NCLIV_020480	Spliceosome component Intron-exon recognition	[24,35]
RNA binding motif 28 protein	NCLIV_033040	Common constituent of snrps	[36]
Sm protein	NCLIV_038830	Core component of spliceosome	[37]
pre-mRNA-splicing factor	NCLIV_064230	Associated to RNA polymerase II	[38]
snrpU3	NCLIV_055310	rRNA splicing	[39]
*D. rerio* Splicing factor 1	F1QP64_DANRE	pre mRNA splicing; recognition of branch site	UP
PRP40	Q7ZUE4_DANRE	mRNA splicing (major pathway)	UP
Sm D1	Q8JHH1_DANRE	Core component of spliceosome	[37]
Protein 24a	A0A0R4ICY2_DANRE	Key regulator of splicing	[40]
snrpU2	Q8JHJ3_DANRE	Spliceosome component	[24]
Protein 8A	RBM8A_DANRE	Pre-mRNA splicing	UP
snrpU1	E7F071_DANRE Q6NUT5_DANRE	Spliceosome component Intron-exon recognition	[24,35]
Poly-U-binding-splicing factor a	F1QZ57_DANRE	Control of pre-mRNA splicing	[41]
RNA-binding motif protein 39b	Q58ER0_DANRE	Regulation of splicing	[42]
Splicing factor 3B subunit 3	SF3B3_DANRE	Assembly of Complex A of the spliceosome	UP
Spliceosome-associated factor 1	Q6DG11_DANRE	Recruiter of spliceosome components U4,U5,U6	UP
Heterogeneous nuclear ribonucleoprotein K	A0A2R8Q139_ DANRE	Poly-C binding, splicing Control of transcription	UP

## Data Availability

Data are made available as Appendix A datasets (see above).

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
