# Peer review of "Common Molecular Targets of a Quinolone Based Bumped Kinase Inhibitor in Neospora caninum and Danio rerio"

_ijms, 2022, doi:10.3390/ijms23042381_

Round 1

Reviewer 1 Report

The Manuscript No “ijms-1603266” entitled “Common molecular targets of a quinolone based bumped kinase inhibitor in Neospora caninum and Danio rerio”.

The manuscript is well designed and written. It provides an interesting finding in the field of molecular biology, pharmacology, parasitology, and could be interesting for a wide section of readers. It could be accepted after minor corrections.

Italicized the Latin names (line 109), and check throughout the MS.

Figure 2 has low resolution could be replaced with more than 300 dpi figure.

I highly encourage the insertion of supplementary figures S1-S4 into the main manuscript, due to its importance in increasing the clarity of the obtained findings.

This study could have Institutional Review Board Statement approval.

According to the author's guidelines could you rearrange the MS sections.

Author Response

Many thanks for your very positive review. Here are our corrections and remarks.

Italicized the Latin names (line 109), and check throughout the MS.

Response: Corrections were made throughout the script.

Figure 2 has low resolution could be replaced with more than 300 dpi figure.

Response: The figure has been resized.

I highly encourage the insertion of supplementary figures S1-S4 into the main manuscript, due to its importance in increasing the clarity of the obtained findings.

Response: The supplemental figures have been included now into the main manuscript (new Figures 2 – 5). The other figures have been renumbered accordingly.

This study could have Institutional Review Board Statement approval.

Response: The statement has been included now.

According to the author's guidelines could you rearrange the MS sections.

Response: The MS sections follow the author`s guidelines, now.

Reviewer 2 Report

  1. In Figure S3, it mentioned “Thicker arrows indicate malformation of the eye…”, I would suggest authors add some description or hypothesis in the discussion section to describe the possible pathway that possibly induced the malformation of the eye in BKI-treated embryos. On the other hand, although BKI-1748 is a quinine-derived compound, the 13 different eluted proteins (shown in Figure 3) indicate these two compounds (BKI-1748 and quinine) exhibit some different physiological effects in zebrafish. Does quinine induce the same malformation of the eye of other organs that like BKI-1748 treatment?
  2. Figure legend of Figure S3. Two dots were shown at the end of the first sentence. In the fourth line, please add “(E-H)” to the end of the sentence (…bursting of the thorax (E-H).
  3. Minor comments: please recheck the journal names that showed in the reference list. Reference no. 10, 17, 19, 27, 30, and 44. The name of journals should be shown in ISO abbreviation style and consistency to other cited references.

Author Response

Thank you for your comments. Please find our responses:

In Figure S3, it mentioned “Thicker arrows indicate malformation of the eye…”, I would suggest authors add some description or hypothesis in the discussion section to describe the possible pathway that possibly induced the malformation of the eye in BKI-treated embryos.

Response: We think that interference with RNA-splicing is the major mechanism responsible for the phenotypes both in N. caninumand zebrafish (as well as in mouse) embryos. See lanes 313-316f in our discussion. However, this is speculation and would need to be further investigated.

On the other hand, although BKI-1748 is a quinine-derived compound, the 13 different eluted proteins (shown in Figure 3) indicate these two compounds (BKI-1748 and quinine) exhibit some different physiological effects in zebrafish. Does quinine induce the same malformation of the eye of other organs that like BKI-1748 treatment?

Response: Since pleiotropic effects of quinine are well known, in particular its effects on embryo development, we firmly believe that this compound will cause similar, if not stronger, effects on zebrafish embryos. Since quinine is, however, out of scope for the treatment of neosporosis and related parasitoses (due to its side effects), we did not include it in our zebrafish embryo study. The only reason for including this compound into our affinity chromatography study was to eliminate BKI-1748 binding proteins binding to the quinoline moiety present in both compounds (see introduction ll.83ff and Figure 1).

Figure legend of Figure S3. Two dots were shown at the end of the first sentence. In the fourth line, please add “(E-H)” to the end of the sentence (…bursting of the thorax (E-H).

Response: The legend has been corrected accordingly.

Minor comments: please recheck the journal names that showed in the reference list. Reference no. 10, 17, 19, 27, 30, and 44. The name of journals should be shown in ISO abbreviation style and consistency to other cited references.

Response: The references have been edited in ISO abbreviation style.